# An Investigation of the Effect of Water Additives on Broiler Growth and the Caecal Microbiota at Harvest

**DOI:** 10.3390/pathogens11080932

**Published:** 2022-08-18

**Authors:** Genevieve Greene, Leonard Koolman, Paul Whyte, Catherine M. Burgess, Helen Lynch, Aidan Coffey, Brigid Lucey, Lisa O’Connor, Declan Bolton

**Affiliations:** 1Teagasc Food Research Centre, Ashtown, Dublin 15, D15 DY05 Dublin, Ireland; 2School of Veterinary Medicine, University College Dublin, Belfield, Dublin 4, D04 V1W8 Dublin, Ireland; 3Department of Agriculture, Food and the Marine, Backweston, Celbridge, W23 X3PH Kildare, Ireland; 4Department of Biological Sciences, Munster Technological University, T12 P928 Cork, Ireland; 5Food Safety Authority of Ireland, George’s Dock, Dublin 1, D01 P2V6 Dublin, Ireland

**Keywords:** broiler, water additives, organic acids, essential oils, medium-chain fatty acids, broiler performance, caecal microbiota

## Abstract

*Campylobacter* is the most common foodborne pathogen in developed countries and most cases are associated with poultry. This study investigated the effect of three anti-*Campylobacter* water additives on broiler growth and on the caecal microbiota at harvest using 16S rRNA amplicon sequencing. Mixtures of organic acids (OA) and essential oils (EO) were administered to broilers for the entirety of the production cycle (35 d) and medium-chain fatty acids (MCFA) for 5 d immediately before harvest, under commercial conditions. Bird weight gain was significantly (*p* < 0.001) reduced in broilers receiving the OA and EO treatments. While this was most likely due to reduced water intake and corresponding lower feed consumption, changes to the caecal microbiota may also have contributed. Firmicutes made up over 75% of the bacteria regardless of sample type, while the minor phyla included Bacteroidetes, Actinobacteria, Melainabacteria, and Proteobacteria. There were no significant (*p* > 0.05) differences in the alpha diversity as measured using ACE, Chao1, and Shannon indices, except for control (water) versus MCFA and OA versus MCFA, using the Wilcox test. In contrast, there was a significant (*p* < 0.05) difference in beta diversity when the treated were compared to the untreated control and main flock samples, while linear discriminant analysis effect size (LeFSe) identified three OTUs that were present in the control but absent in the treated birds. It was concluded that the water additives tested adversely affected broiler performance, which may, at least in part, be due to changes in the caecal microbiota, assuming that the altered microbiota at day 35 is indicative of a change throughout the production cycle.

## 1. Introduction

*Campylobacter* spp. cause approximately 250,000 cases of gastroenteritis in the EU every year, costing an estimated €2.4 bn in healthcare and lost working days [1,2,3,4]. Although ubiquitous in warm-blooded animals, the primary reservoir is in birds, and broilers are the main source of *Campylobacter* infections in humans [5].

Water additives could potentially be used to control *Campylobacter* in broilers. Indeed, organic acids (OA), essential oils (EO), and medium-chain fatty acids (MFCA), which are amongst the most promising anti-*Campylobacter* water additives, may also promote gut health by increasing the quantity and diversity of probiotic bacteria such as *Lactobacillus* and *Bifidobacterium* [6,7].

Organic acid molecules, in the undissociated form, pass through the bacterial cell membrane and dissociate into charged anions and protons. Thus, the cellular hydrogen ion equilibrium is disrupted, causing an increase in the pH, the inhibition of essential metabolic reactions, and the accumulation of toxic anions [8]. Byrd et al. [9] observed reduced *Campylobacter* prevalence in broilers consuming water which contained a low (0.44%, *v*/*v*) concentration of lactic acid. Essential oils are phenolic compounds capable of causing structural breakdown and altered permeability of the bacterial cell membrane, resulting in cellular leakage and death, and have potent anti-*Campylobacter* activity [10]. Carvacrol treatment has also been associated with reduced cell motility in *Campylobacter* [11]. The mode of action of MCFAs is not completely understood but they may act as non-ionic surfactants, embedding themselves in the bacterial lipid bilayer and forming pores, resulting in cell leakage and destruction [8]. MCFAs have previously been reported to significantly reduce *Campylobacter* carriage in broilers when administered as a water or feed additive [4,12].

While OA and EO are readily consumed in water by broilers, previous (unpublished) research in our group found that MFCAs were less palatable and would only be consumed for a few days. However, these water additives may alter the broiler gut microbiota, especially in the caeca, where up to 10% of a broilers’ metabolisable energy is generated, and thus may adversely affect broiler performance [3,4,5,6,7,8,9,10,11,12,13,14,15,16,17].

The link between a microbially diverse healthy gut microbiota and optimal broiler growth is well established [18]. Bacteria in the gut have an important role in intestinal morphology, nutrient digestion, nutrient absorption, and host health [6,19]. In a healthy established microbial community, at least 85–90% of bacteria present are beneficial Gram-positive bacteria such as *Lactobacillus*, *Eubacterium*, *Streptococcus*, and *Bifidobacterium* [20,21]. The remaining 10–15% of bacteria include *Bacteroides*, *Prevotella*, *Fusobacterium*, *Selenimonas*, and *Megasphaera,* with *Clostridium* also present in younger birds [20,21].

The caecum is the most densely populated section of the broiler GIT and more than 90% of the bacteria present are Gram-positive, with Firmicutes, Bacteroidetes, and Actinobacteria being the most dominant phyla. It is also the primary colonisation site for pathogens such as *Campylobacter* and hence the target for water additive pathogen control treatments. Despite the potential application of OAs, MCFAs, and EOs as anti-*Campylobacter* water additives in broiler production, research is required on their effect on the caecal microbiota and broiler performance before commercial application [7]. The objective of the present study was to investigate the effect of OA-, EO-, and MCFA-based anti-*Campylobacter* water additives on broiler performance and on the caecal microbiota using 16S rRNA amplicon sequencing.

## 2. Results

### 2.1. The Effect of OA and EO Water Additives on Broiler Performance

Broilers receiving the OA and EO water treatments had significantly (*p* < 0.05) decreased body weight gain (approximately half) as compared to the birds in the main flock. The body weight gain of birds receiving the MCFA treatment, which was only administered for the 5 days immediately before harvest, was not affected when compared to the general flock (Table 1).

### 2.2. Most Abundant Taxa in Each Group

The microbiota from 30 caecal samples (five from the birds treated with each of OA, MFCA, and EO, five from the control (plain water) and 10 from the main flock) were analysed using 16S rRNA amplicon sequencing. A mean of 56,939 effective tags were obtained per sample, with a mean length of 411 nucleotides per sequence obtained. Samples were then classified into OTUs based on the 97% identity level, giving a mean of 585 OTUs per sample. The 10 most abundant taxa of in each group were identified. The relative abundance of each taxa at the phylum level is shown in Figure 1.

### 2.3. Most Abundant Phyla and Genera in Each Group

The most abundant phylum, regardless of treatment, was Firmicutes, which made up over 75% of the bacteria in all groups. Bacteroidetes was the second most prevalent phylum (8% to 10%) and the Firmicutes:Bacteroidetes ratio (F:B) was significantly (*p* < 0.0001) different in the OA- and MCFA-treated groups versus the control and the main flock. The relative abundance of the minor phyla varied depending on the treatment group: control: Actinobacteria (3%) > Proteobacteria (2.9%) > Melainabacteria (1.2%); main flock: Actinobacteria (5.8%) > Proteobacteria (1.4%) > Melainabacteria (0.7%); OA: Proteobacteria (5.5%) > Melainabacteria (3.9%) > Actinobacteria (3.0%); EO: Proteobacteria (3.3%) > Actinobacteria (0.9%) > Melainabacteria (0.5%), and MCFA: Actinobacteria (8.9%) > Proteobacteria (1.5%) > Melainabacteria (0.9%).

Data on the most abundant genera are provided in Figure 2, which shows the 35 most prevalent genera in each group. In the main flock, these included Flavonifractor, Parabacteroides, Shuttleworthia, Eisenbergiella, unidentified Lachnospiraceae, unidentified Cyanobacteria, and Erysipelatoclostridium, while Streptococcus, Ruminiclostridium, Fecalibacterium, and Oscillibacter were most prevalent in the control group. In the treated groups, Tyzzerella, Comamonas, Intestinimonas, Butyricicoccus, Alistipes, Bilophilia, unidentified Melainabacteria, Lachnoclostridium, Parasutterella, unidentified Clostridiales, Blautia, and Subdoligranium were the most common genera detected when OA was administered to the birds; Tyzzerella, Comamonas, Intestinimonas, Butyricicoccus, Alistipes, Bilophilia, unidentified Melainabacteria, Lachnoclostridium, Parasutterella, unidentified Clostridiales, Blautia, and Subdoligranium after EO, and Anarostipes, Faecalibacterium, Bifidobacterium, Bacteroides, and Caproiciproducens following MCFA application.

### 2.4. Shared and Unshared OTUs

The number of OTUs that were common and exclusive to each of the treatment caeca and the control group is shown in Figure 3A. A total of 571 OTUs were shared between all four groups, and each treatment group had 111–189 unshared OTUs, with the OA group having the largest amount of unshared OTUs at 189. Figure 3B shows the OTUs that were shared and unshared between treatment groups and the general flock. There was a total of 588 shared OTUs between the four groups, with each individual group having between 91 and 193 unshared OTUs. The general flock had the greatest amount of unshared OTUs, with 193.

### 2.5. Diversity

Alpha diversity measures how much diversity there is within each sample, while beta diversity measures the (dis)similarity between samples (i.e., it quantifies differences in the overall taxonomic composition between two samples).

#### 2.5.1. Alpha Diversity

There was broad agreement between the ACE, Chao1, and Shannon diversity indices (Figure 4A–C, respectively), based on the OTU profiles. There was no significant (*p* > 0.05) difference in alpha diversity regardless of index used, with the exception of control versus MCFA, and OA versus MCFA, when the Wilcox test was used. Thus, based on analysis of the OTUs, the OA and EO treatments did not result in lower microbial diversity in the caeca, while the diversity of the bacteria in the caeca of the MCFA-treated birds was significantly (*p* < 0.05) lower.

#### 2.5.2. Beta Diversity

Beta indices describe compositional changes between different microbial communities. There was a significant (*p <* 0.05) difference in beta diversity when the treated were compared to the untreated control and main flock samples. The results are visualised in the PCoA plot (Figure 5).

A Linear Discriminant Analysis (LDA) histogram showing the OTUs that account for the differences between treatment groups is provided in Figure 6. The general flock and the control group have four and three OTUs, respectively, which differentiate them from the treatment groups. The OTUs differentiating the general flock are associated with the *Lactobacillus salivarius*, Pantoea, Tannerellaceae, and Parabacteroides, while OTUs specific to the control group belong to the Streptococcus, Streptococcaceae, and *Streptococcus gallolyticus* subsp. *macedonicus*. The OTUs differentiating the OA treatment group from the remainder of the treatment groups included Proteobacteria, Butyricicoccus, Comamonas, and Comamonas kerstesii taxa, while Lactobacillales, Bacilli, Lactobacillaceae, Lactobacillus, *Lactobacillus reuteri*, *Methylobacterium fujisawaense*, Methylobacterium, and Beijerinckiaceae differentiated the EO treatment group. There were five OTUs that were only found in the MCFA treatment group, including the order Bifidobacteriales (and associated family (Bifidobacteriaceae) and genus (Bifidobacterium)), unidentified Actinobacteria, and Fecalibacterium taxa.

## 3. Discussion

Birds receiving the OA or EO mixture in their drinking water developed more slowly, resulting in 47% and 54% less weight gain, respectively, as compared to the general flock after 35 days, where the weight gains were similar to those observed in birds receiving the water control and MCFAs. The latter observation with MCFA treatment was most likely due to the late administration (last 5 days before harvest) of this treatment. While the most likely cause of the reduced growth in birds receiving OA and EO treated water was reduced water intake driving lower feed consumption [22], changes in the microbiota may also have contributed to the reduced broiler performance. The Firmicutes:Bacteroidetes ratio (F:B), for example, was significantly (*p <* 0.0001) higher in the OA treated birds versus the untreated birds, and previous studies have suggested that a higher F:B ratio is associated with lower growth performance in broilers [17,23]. Moreover, Lachnospiraceae and Eisenbergiella, which were more abundant in the untreated birds, are essential for efficient energy, amino acid, and nucleotide metabolism [24] and have been previously associated with high-performing birds [23]. In the EO-treated birds, this situation may have been compounded by the fact that the Lachnospiraceae were mainly displaced by Enterobacteriaceae, higher gut concentrations of which are associated with reduced broiler performance [17]. However, not all studies are in agreement. It might have been expected that the statistically significant (*p* < 0.05) higher abundance of Lactobacillus in the EO-treated broilers and Actinobacteria (mainly Bifidobacterium) in the MCFA group would have resulted in improved broiler performance. Lactobacillus species have been shown to stimulate multiple aspects of the immune response and can even prevent pathogen colonisation in chickens [25,26]. Bifidobacterium stabilise the gastrointestinal barrier, modulate local and systemic immune responses, inhibit the cellular invasion of pathogens, and promote the bioconversion of unavailable dietary compounds into bioactive healthy molecules [7]. Thus, our contradictory findings suggest that the interaction between the microbiota and broiler performance is complex, and the positive impact of increasing a beneficial bacterial population can be offset by other changes in the microbiota.

Firmicutes, at a concentration of over 75%, were the dominant phylum in all of our caecal samples, followed by Bacteroidetes at concentrations which ranged from 8% to 10%, as widely reported in other broiler caecal microbiota studies [27,28,29,30]. The third most common phylum was Actinobacteria in the control birds (3%), the main flock (5.8%), and the MCFA group (8.9%), and Proteobacteria in the OA- (5.5%) and EO-treated (3.3%) birds. Actinobacteria are usually the third most prevalent phylum in the broiler GIT, followed by Proteobacteria [31]. Interestingly, Melainabacteria made up 3.9% of the OA samples as compared to 1.2%, 0.7%, 0.5%, and 0.9% in the control, main flock, EO, and MCFA samples, respectively. The higher concentration of Melainabacteria in the OA group may be due to their ability to utilise organic acids as a carbon source [32]. The relatively high abundance of Actinobacteria (8.9%) in the MCFA treatment group has been previously reported but remains unexplained [33].

With the exception of the MCFA treatment, the alpha diversity was not significantly (*p* > 0.05) affected by the water additives. Cuccato et al. [34] reported similar findings when studying the effects of antimicrobial treatments on the microbiota in broiler caeca and suggested that the high microbial diversity in the caecum could mask the impact of antimicrobial treatments on the alpha diversity. However, the beta diversity analysis suggested that the diversity within the same set of treated samples (e.g., the five OA samples or the five EO samples or the five MCFA samples) was dissimilar. Thus, the effect of a given water additive on the microbial composition within the caeca of a given bird was not consistent across all five birds tested in this study.

## 4. Materials and Methods

### 4.1. Description of Farm Conditions

This study was performed in one broiler house on a commercial poultry farm in county Monaghan in the Republic of Ireland. Prior to chick placement, pens were set up for each of the three treatments plus a control (water). Each pen was constructed from 4 galvanised steel mesh panels and had a total of 8.4 m^2^ floor space divided into 4 equal areas, as shown in Figure 7. The perimeter of each pen was surrounded with cardboard to prevent the chicks from escaping in the first 2 weeks of production.

### 4.2. Animals, Treatment, and Management

A total of 208 one-day-old *Gallus gallus domesticus* (Ross breed) broiler chicks were used in this study. On the day of chick placement, each pen was populated at a density of 52 birds, 13 bird per sub-pen. Birds placed in each pen received one of three treatments (OA, EO, or MCFA) or a control (untreated water). All treatments were provided as water additives *ad libitum* using 3L bird drinkers (McCabes General Merchants, Cavan, Ireland), refilled when necessary. Test birds were raised under the same conditions as the rest of the main flock, receiving the same food, litter, and other environmental conditions. The OA solution consisted of lactic acid (W261106-1KG-K) (1.25%, *v*/*v*) and potassium sorbate (85520) (1.5%, *w*/*v*), the EO solution consisted of carvacrol (W224502) 0.125%, *v*/*v* and thymol (T0501) (0.25%, *w*/*v*), while the MCFA was sodium caprylate (C5038) (1.5%, *w*/*v*). All chemicals were supplied by Sigma Aldrich (Wicklow, Ireland). The OA and EO solutions were administered in the water throughout the production cycle, while the MCFA solution was provided to the birds from day 30 to 35.

### 4.3. Sample Collection and Preparation

At 35 days old (one day before harvest of the flock), 5 birds from each of the test treatments and the control group (one from each of the 4 sub-pens per treatment and one chosen at random) and 10 birds randomly selected from the flock were removed (*n* = 30), transported to a local veterinary practice, euthanised by a veterinarian using cervical dislocation, and had their gastrointestinal tracts (GIT) aseptically removed. These GIT samples were then transported to our laboratory in a cool box (2 to 4 °C), where the caecal contents were aseptically removed and stored at −70 ℃.

Throughout the broiler rearing period, the weights of birds were continuously monitored. On days 1, 4, 7, 11, 14, 18, 21, 25, 28, 32, and 35, the weights of 2 birds in each of the smaller pens and 5 birds from the flock were measured using the BW-2050 weight system (Weltech International Limited, Cambridgeshire, UK).

### 4.4. DNA Extraction and 16S rRNA Sequencing

DNA was extracted from the 30 caecal content samples using the DNeasy PowerSoil Pro Kit (Qiagen, Manchester, UK), following the manufacturer’s protocol. DNA concentrations were measured with a NanoDrop spectrophotometer (NanoDrop 1000, ThermoFisher Scientific, Dublin, Ireland). DNA purity was assessed on a 1% agarose gel and diluted, in sterile water, to a concentration of 1 ng/μL, and samples were sent to Novogene Bioinformatics Technology Co., Ltd. (Beijing, China) for 16S amplicon metagenomics sequencing.

Bacterial DNA specific to the V3-V4 region of the 16S rRNA gene was amplified using the primer pair 341F (CCTAYGGGRBGCASCAG) and 806R (GGACTACNNGGGTATCTAAT) along with a barcode [35]. PCR reactions were performed with Phusion^®^ High-Fidelity PCR Master Mix (New England Biolabs, Ipswich, MA, USA). Following this, PCR products underwent quality control and quantification checks. PCR products were electrophoresed on a 2% agarose gel, and those bands with a bright band between 400 and 450 bp were excised using the Qiagen Gel Extraction Kit (Qiagen, Manchester, UK). DNA libraries were prepared using the NEBNext^®^ Ultra ^TM^ DNA Library Prep Kit for Illumina and quantified via Qubit and qPCR, and analysed using the Illumina NovaSeq 6000 platform.

### 4.5. Statistical Analysis

#### 4.5.1. Sequence Data Processing

Paired-end reads, with an approximate length of 400 base pairs, were assigned to samples based on unique barcodes and truncated by removing the barcode and primer sequences, merged using the FLASH (V1.2.7) analysis tool [36], and the splicing sequences were termed “raw tags”. Quality filtering of raw tags was performed under specific filtering conditions, obtaining high-quality clean tags according to the Quantitative Insights into Microbial Ecology (QIIME, V1.7.0) pipeline quality-controlled process [37,38]. Raw tags were compared with the reference in the SILVA (release 138) database using the UCHIME algorithm to detect chimera sequences, which were subsequently removed, with the effective tags then obtained [39,40].

#### 4.5.2. OTU Cluster and Taxonomic Annotation

Sequence analysis on all effective tags was performed using Uparse software (Uparse v7.0.1090). Sequences with ≥97% similarity were assigned to the same Operational Taxonomic Unit (OTU) [41]. A representative sequence for each OTU was screened for further annotation as follows. Representative sequences were run through QIIME (Version 1.7.0) and Mothur pipelines using the SILVA SSUrRNA (release 138) database for species annotation at the taxonomic level (kingdom, phylum, class, order, family, genus, species) [42,43,44]. The phylogenetic relationships between the representative sequences of all OTUs were obtained using MUSCLE (Version 3.8.31) [45]. OTU abundance data were normalised using a standard, with the sequence number corresponding to the sample with the fewest sequences, with alpha and beta diversity analysis performed on this normalised data.

#### 4.5.3. Alpha and Beta Diversity Measurements

Alpha diversity measurements were used to analyse the biodiversity of each sample using 3 indices: ACE, Chao1, and Shannon. All indices were calculated with QIIME (Version 1.7.0) and displayed via R software (Version 2.15.3). The *p* values, for significance, were calculated via a permutation test (Tukey). The *t*-test and drawing were conducted by R software.

Beta diversity analysis was applied on both weighted and unweighted UniFrac to evaluate the sample variances in species complexity using QIIME software. Cluster analysis and Principal Component Analysis (PCA) were then performed using R software with FactoMineR and ggplot2 packages to reduce the dimensions of original variables. To establish principal coordinates and visuals from complex multidimensional data, Principal Coordinate Analysis (PCoA) was performed. Thereafter, a distance matrix of the weighted UniFrac among samples was transformed to a new set of orthogonal axes, where the maximum variation factor was demonstrated by the first principal coordinate, and the second maximum variation factor was demonstrated by the second principal coordinate. PCoA analysis was displayed using the WGCNA, stat, and ggplot2 packages in R software (Version 2.15.3).

#### 4.5.4. LEfSe Analysis

LEfSe (Linear Discriminant Analysis Effect Size) analysis was used to determine the OTUs that most likely explained the differences between samples and was conducted using LefSe software, Boston, MA, USA.

#### 4.5.5. Analysis of Bird Weights

Bird weights were analysed using ANOVA, using Graphpad Prism ver. 7.2 (Graphpad Software Incorporated, San Diego, CA, USA).

## 5. Conclusions

It was concluded that the water additives tested adversely affected broiler performance, which may, at least in part, be due to changes in the caecal microbiota. Thus, future studies aimed at developing anti-*Campylobacter* treatments should include an assessment of the impact on gut microbiota and broiler performance in addition to pathogen reduction in the caeca. Moreover, our understanding of the link between the caecal microbiota and broiler performance would be enhanced if periodic samples were obtained and analysed to provide information on the dynamics of any changes with respect to time/age of the birds and feed type.

## Figures and Tables

**Figure 1 pathogens-11-00932-f001:**
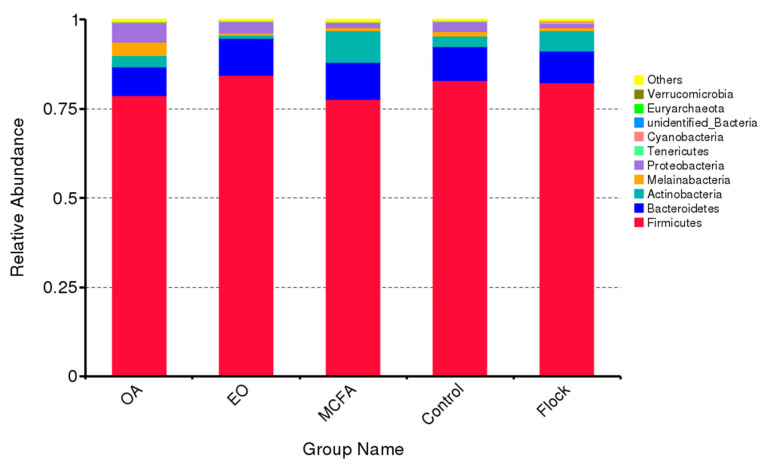
The 10 most abundant taxa in the caeca from the groups treated with OA, EO, MCFA, water (control), and the main flock.

**Figure 2 pathogens-11-00932-f002:**
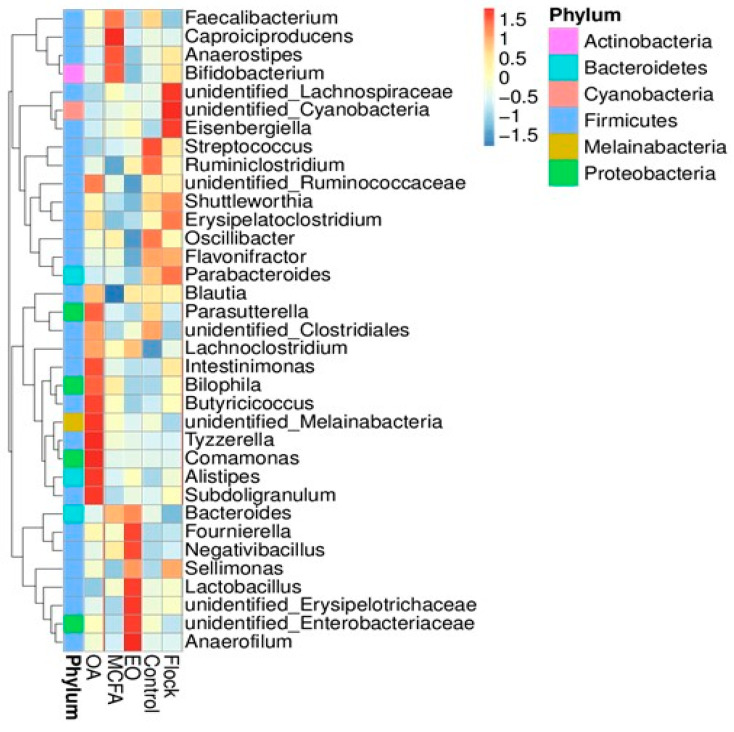
Heat map illustrating the 35 most frequent genera in the caeca of each group.

**Figure 3 pathogens-11-00932-f003:**
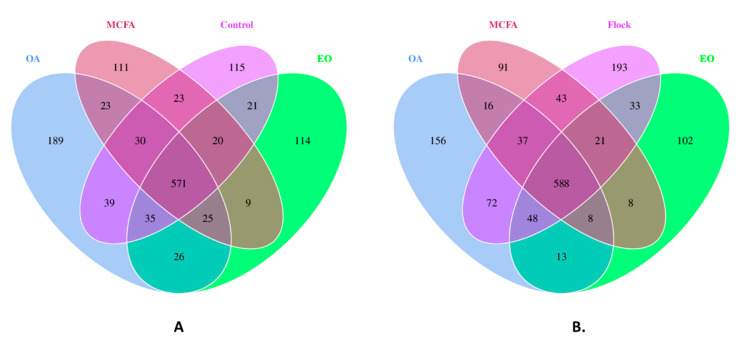
(**A**) Shared and unshared OTUs between the OA, MCFA, and EO treatment groups and the control group and (**B**) the OTUs that are shared and unshared between the OA, MCFA, and EO treatment groups and the general flock.

**Figure 4 pathogens-11-00932-f004:**
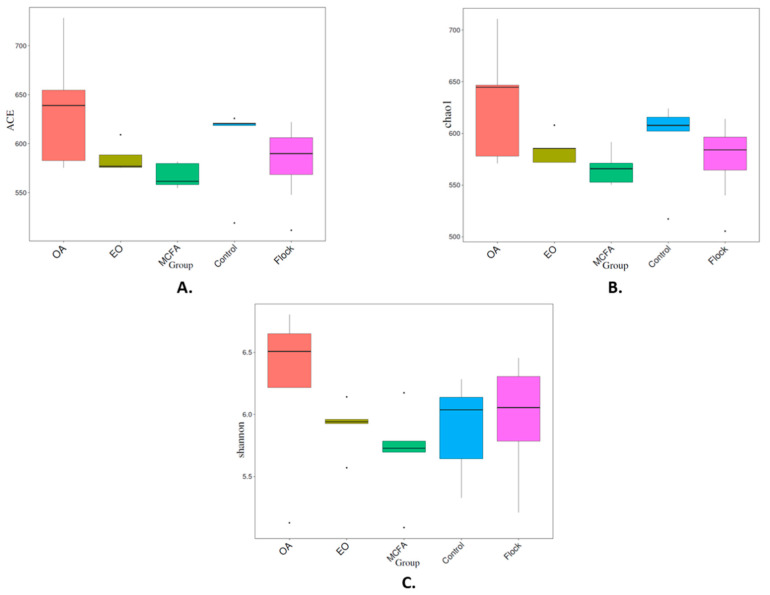
Comparison of the alpha diversity of the different groups using (**A**) ACE index; (**B**) Chao1 index; (**C**) Shannon index, based on the OTU profile.

**Figure 5 pathogens-11-00932-f005:**
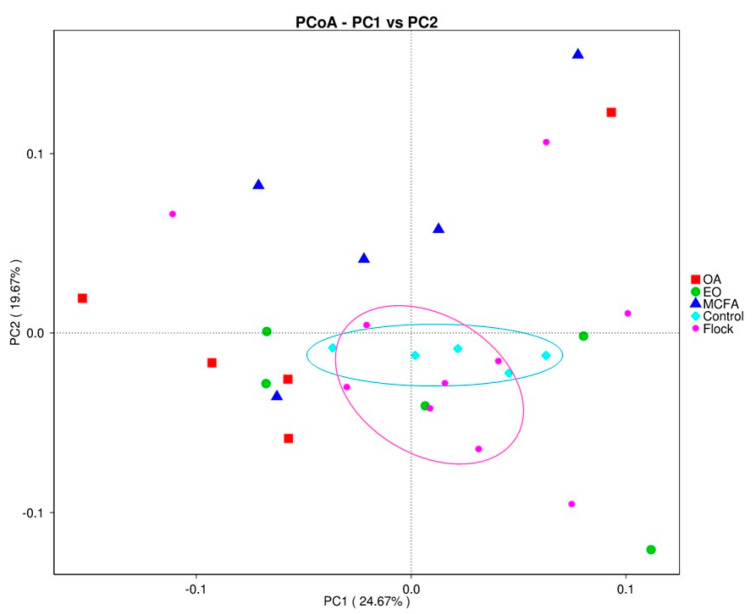
Principal Coordinate Analysis plot showing the similarity and dissimilarity between samples in each treatment group.

**Figure 6 pathogens-11-00932-f006:**
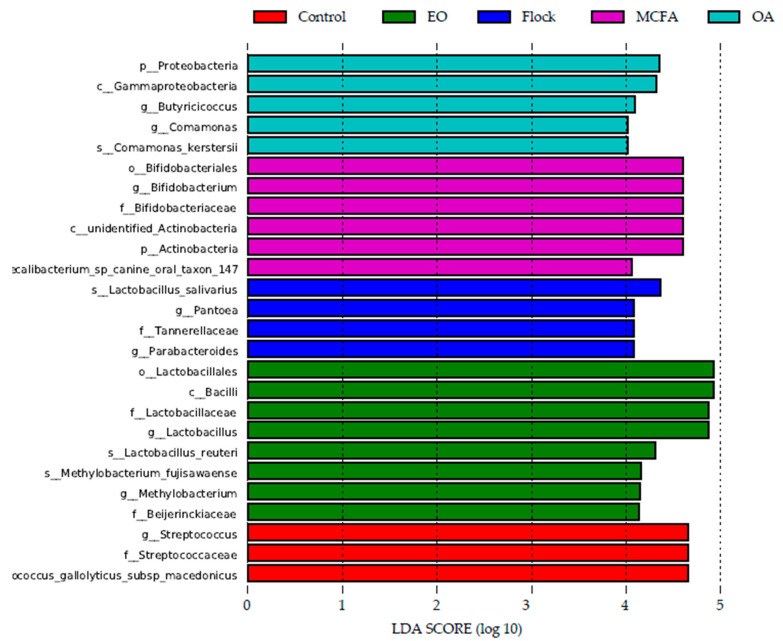
LDA histogram depicting the OTUs accounting for the differences observed between treatment groups as identified by the LEfSe algorithm.

**Figure 7 pathogens-11-00932-f007:**
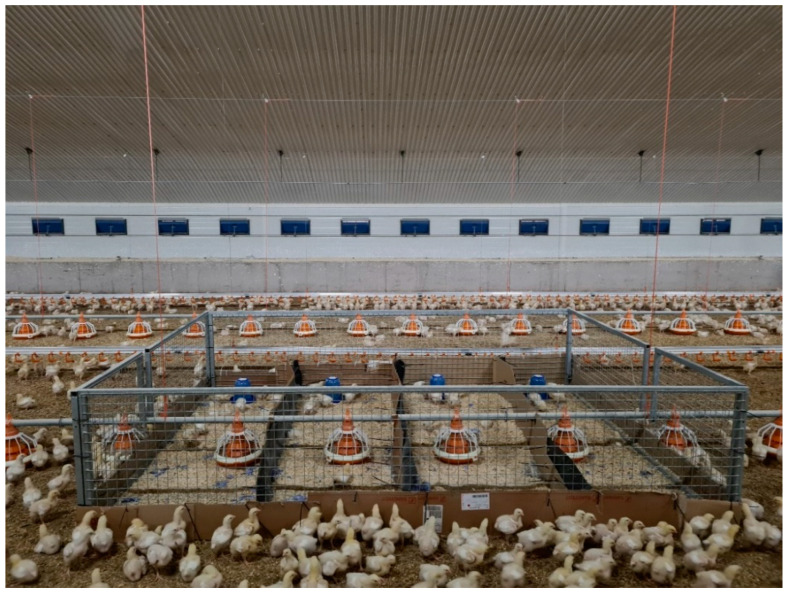
The type of bird pen used in this study.

**Table 1 pathogens-11-00932-t001:** The mean weight of broilers during the production cycle when administered OA (1.25% *v*/*v* lactic acid and 1.5% *wt*/*v* potassium sorbate), EO (0.25% *wt*/*v* thymol and 0.125% *v*/*v* carvacrol), and MCFA (1.5% *wt*/*v* sodium caprylate) treatments in comparison to the control and the main flock.

	Mean Weight ± Standard Error (Kg)				
Time (d)	OA	EO	MCFA	Control (Water)	Main Flock
1	0.045 ± 0.007	0.051 ± 0.003	0.051 ± 0.003	0.056 ± 0.003	0.055 ± 0
4	0.08 ± 0.001	0.091 ± 0.002	0.105 ± 0.001	0.111 ± 0.002	0.109 ± 0
7	0.121 ± 0.009	0.161 ± 0.009	0.193 ± 0.007	0.173 ± 0.018	0.183 ± 0
11	0.169 ± 0.006	0.299 ± 0.017	0.355 ± 0.019	0.379 ± 0.009	0.35 ± 0.019
14	0.265 ± 0.013	0.441 ± 0.015	0.541 ± 0.028	0.584 ± 0.004	0.591 ± 0.024
18	0.324 ± 0.022	0.545 ± 0.04	0.728 ± 0.024	0.822 ± 0.041	0.827 ± 0.062
21	0.445 ± 0.042	0.713 ± 0.064	1.001 ± 0.034	1.015 ± 0.037	0.897 ± 0.042
25	0.612 ± 0.026	0.922 ± 0.098	1.478 ± 0.07	1.568 ± 0.054	1.391 ± 0.08
28	0.826 ± 0.045	1.008 ± 0.104	1.707 ± 0.127	1.898 ± 0.086	1.66 ± 0.127
32	0.933 ± 0.106	1.102 ± 0.067	2.019 ± 0.096	2.19 ± 0.136	2.08 ± 0.017
35	1.087 ± 0.09	0.952 ± 0.049	2.324 ± 0.139	2.616 ± 0.085	2.057 ± 0.106
*p* value ^1^	<0.0001	<0.0001	0.9062	0.0358	N/A

^1^ compared to main flock.

## Data Availability

The raw read sequence data are available at NCBI SRA Bioproject PRJNA847752.

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
