# Peer review of "An Investigation of the Effect of Water Additives on Broiler Growth and the Caecal Microbiota at Harvest"

_pathogens, 2022, doi:10.3390/pathogens11080932_

Round 1

Reviewer 1 Report

Overall a well designed study  which assess the impact of additives given orally via water on the caecal microbiome and broiler performance. The data has been presented well with suitable discussions around the findings. Just one thing that is unclear and that I would suggest the authors add into the introduction is why MCFA are given later in life, this information would greatly benefit readers

Author Response

Overall a well designed study, which assess the impact of additives given orally via water on the caecal microbiome and broiler performance. The data has been presented well with suitable discussions around the findings. Just one thing that is unclear and that I would suggest the authors add into the introduction is why MCFA are given later in life, this information would greatly benefit readers.

Response: An explanation has been provided in lines 60-62.

Thank you for your review.

Reviewer 2 Report

The authors have conducted a study on the effect of water additives on broiler performance and caecal microbiota. This is an interesting subject. However, the microbiota was only analysed on 5 samples of each treatment on broilers of 35 days old. No data are shown on microbiota during the production cycle. The OA and EO treatments were administered throughout the production cycle, while MCFA was administered only from day30 and 35. In order to evaluate the effect on caecal microbiota more data are needed.

Author Response

The authors have conducted a study on the effect of water additives on broiler performance and caecal microbiota. This is an interesting subject. However, the microbiota was only analysed on 5 samples of each treatment on broilers of 35 days old. No data are shown on microbiota during the production cycle. The OA and EO treatments were administered throughout the production cycle, while MCFA was administered only from day30 and 35. In order to evaluate the effect on caecal microbiota more data are needed.

Response: I agree, the title is misleading and has been revised to better reflect the objective of the study. More frequent sampling times would provide additional data and would describe changes in the caecal microbiota throughout production.  However, this would be limited to OA and EO as the MCFA were only administered for the last 5 days. Regardless, this study provides the basis for an additional funding application to investigate the microbiota at different stages throughout the growth cycle.  Our conclusion, in this paper is therefore based on the assumption that the altered microbiota on day 35 is indicative of changes throughout broiler growth and this has been indicated in the text where required.

Thank you for your review.

Reviewer 3 Report

This study focused on the impact of water additives on broiler performance and caecal microbiota change and evaluated the function of three anti-Campylobacter additives based on 16S rRNA sequencing. The weight gain of broilers fed organic acids (OA) and essential oils (EO) was significantly (P<0.001) reduced in the treatment group, and some clues were obtained in the following 16S rRNA sequencing results. It concludes that the tested water additives adversely affected the performance of broilers, which may be due at least in part to changes in the type and number of caecal microbiota. There are still some problems to be solved in this paper. If the following problems can be solved well, this paper is significant for developing anti-Campylobacter water additives. Details are as follows:

1) In the abstract part, the effects of AO, EO and MFCA water additives on Campylobacter were not introduced.

2) For the three substances, AO and EO were administered in one period, while MFCA was administered in another period. The following article lacks a specific explanation. In the case of comparison of the other three substances, how to analyze the results brought by different addictives delivery methods should be discussed

3) The results of 16S rRNA sequencing could not correspond well to the production performance results. However, in the abstract, it was emphasized that the reduced performance of broilers caused by water additives might be partly due to the change in caecal microbiota. Production performance and caecal microflora changes can be concluded separately, or further data analysis can enhance the correlation between production performance and sequencing results.

4) In the Introduction, the background of AO, EO and MFCA water additives is insufficient, which cannot reflect the importance of the fundamental research.

5) Is it insufficient to use only body weight observation data to express broiler performance? Data used to measure changes in production performance, such as Feed Intake (FI) and Feed Conversion Ratio (FCR), were unavailable in this paper.

6) In Table 1, the header of the Table is "Main flock", and the note is "Compared to the general flock". The same group should be written in the same way.

7) For the data related to production performance, the paper does not reflect the method of weight data processing and P-value calculation.

8) There are some formatting and layout problems in the article's chart. In Figures 2-4, the text of the figure is very vague. Figure 6. Should the font be consistent with the full-text diagram? There are two "Figure 6" in the article. The second "Figure 6" should be labelled "Figure 7".

9) Figure 5 shows beta diversity, but the clustering of the Folk group is poor. Therefore, it is not exact to conclude that the untreated samples are separated from the others.

10) The Control and Flock groups in Figure 3 must be represented in different colours.

11) In the last paragraph of the discussion, Cuccato et al. 's results should be discussed after "alpha diversity" and then explained "beta diversity", which would make the article more smooth.

12)Note that the significance P value of the full text should be italicized.

Author Response

This study focused on the impact of water additives on broiler performance and caecal microbiota change and evaluated the function of three anti-Campylobacter additives based on 16S rRNA sequencing. The weight gain of broilers fed organic acids (OA) and essential oils (EO) was significantly (P<0.001) reduced in the treatment group, and some clues were obtained in the following 16S rRNA sequencing results. It concludes that the tested water additives adversely affected the performance of broilers, which may be due at least in part to changes in the type and number of caecal microbiota. There are still some problems to be solved in this paper. If the following problems can be solved well, this paper is significant for developing anti-Campylobacter water additives. Details are as follows:

In the abstract part, the effects of AO, EO and MFCA water additives on Campylobacter were not introduced.

Response: Maybe I misunderstand this comment but I think they are covered in the Abstract so I assume you mean the ‘Introduction’ where they are now described in lines 45-58..

For the three substances, AO and EO were administered in one period, while MFCA was administered in another period. The following article lacks a specific explanation. In the case of comparison of the other three substances, how to analyze the results brought by different additives delivery methods should be discussed

Response: An explanation as to why the MCFAs were not administered until the last 5 days of production has been added in lines 59-61. The possible impact of this delayed administration has now been discussed in lines 205-210.

The results of 16S rRNA sequencing could not correspond well to the production performance results. However, in the abstract, it was emphasized that the reduced performance of broilers caused by water additives might be partly due to the change in caecal microbiota. Production performance and caecal microflora changes can be concluded separately, or further data analysis can enhance the correlation between production performance and sequencing results.

Response: Changes in the microbiota are most likely a minor contributor to the observed redced growth of the broilers with other factors such as reduced water intake and associated feed consumption having a bigger impact. The text has been changed in lines 20-22 and 205-210 to clarify this point. A lot more information/data would be required to establish a relationship between the microbiota and growth in the birds, which is not available in the peer reviewed literature.  This study provides the justification for research to establish this information/research.

In the Introduction, the background of AO, EO and MFCA water additives is insufficient, which cannot reflect the importance of the fundamental research.

Response: As above now described in lines 45-58.

Is it insufficient to use only body weight observation data to express broiler performance? Data used to measure changes in production performance, such as Feed Intake (FI) and Feed Conversion Ratio (FCR), were unavailable in this paper.

Response: We used weight gain as the simplest metric to measure performance. Other metrics such as feed conversion rate could also have been used but in this case were not necessary as the treated birds were clearly not fit for harvest, regardless of whether their feed intake or feed conversion ratios were significantly lower than the control birds.

In Table 1, the header of the Table is "Main flock", and the note is "Compared to the general flock". The same group should be written in the same way.
Response: ‘general’ has been changed to main’.

For the data related to production performance, the paper does not reflect the method of weight data processing and P-value calculation.
Response: this has been corrected, see lines 352-353.

There are some formatting and layout problems in the article's chart. In Figures 2-4, the text of the figure is very vague. Figure 6. Should the font be consistent with the full-text diagram? There are two "Figure 6" in the article. The second "Figure 6" should be labelled "Figure 7".
Response: The Figures have been revised as best we can now. Rather than delay the review process I have resubmitted and will go back to source and revise as suggested in the next draft. The font has been corrected in Figure 6 and the ‘second’ Figure 6 is now Figure 7.

Figure 5 shows beta diversity, but the clustering of the flock group is poor. Therefore, it is not exact to conclude that the untreated samples are separated from the others.
Response: This sentence has been deleted.

The Control and Flock groups in Figure 3 must be represented in different colours.
Response: They are in separated figures and clearly labelled. However, we will go back to the originals and change as suggested.

In the last paragraph of the discussion, Cuccato et al. 's results should be discussed after "alpha diversity" and then explained "beta diversity", which would make the article more smooth.

Response: Changed as suggested.

Note that the significance P value of the full text should be italicized.
Response: Corrected as suggested.

Round 2

Reviewer 2 Report

The authors have made changes in the manuscript., including the title. I consired that tha manuscript is acceptable for publication.